# Diverse Defenses: A Perspective Comparing Dipteran Piwi-piRNA Pathways

**DOI:** 10.3390/cells9102180

**Published:** 2020-09-27

**Authors:** Stephanie Gamez, Satyam Srivastav, Omar S. Akbari, Nelson C. Lau

**Affiliations:** 1Division of Biological Sciences, Section of Cell and Developmental Biology, University of California, San Diego, CA 92093, USA; sgamez@ucsd.edu (S.G.); oakbari@ucsd.edu (O.S.A.); 2Department of Molecular Biology and Genetics, Cornell University, Ithaca, NY 14853-2703, USA; sps257@cornell.edu; 3Department of Biochemistry and Genome Science Institute, Boston University School of Medicine, Boston, MA 02118, USA

**Keywords:** transposons, piRNA, Drosophilids, mosquitoes

## Abstract

Animals face the dual threat of virus infections hijacking cellular function and transposons proliferating in germline genomes. For insects, the deeply conserved RNA interference (RNAi) pathways and other chromatin regulators provide an important line of defense against both viruses and transposons. For example, this innate immune system displays adaptiveness to new invasions by generating cognate small RNAs for targeting gene silencing measures against the viral and genomic intruders. However, within the Dipteran clade of insects, Drosophilid fruit flies and Culicids mosquitoes have evolved several unique mechanistic aspects of their RNAi defenses to combat invading transposons and viruses, with the Piwi-piRNA arm of the RNAi pathways showing the greatest degree of novel evolution. Whereas central features of Piwi-piRNA pathways are conserved between Drosophilids and Culicids, multiple lineage-specific innovations have arisen that may reflect distinct genome composition differences and specific ecological and physiological features dividing these two branches of Dipterans. This perspective review focuses on the most recent findings illuminating the Piwi/piRNA pathway distinctions between fruit flies and mosquitoes, and raises open questions that need to be addressed in order to ameliorate human diseases caused by pathogenic viruses that mosquitoes transmit as vectors.

## 1. Introduction

In the Dipteran clade of insects, arguably the most widely studied groups of species are the Drosophilidae fruit flies and the Culicidae mosquitoes. Whereas *Drosophila* genetics pioneered fundamental basic understanding of development, genetics, genomics, and insights into human disease-causing genes, mosquitoes have been studied intensely because they transmit serious pathogens that affect humans like *Plasmodium falciparum* (malaria parasite), West Nile virus (WNV), chikungunya virus (CHIKV), Yellow fever virus (YFV), dengue (DENV), and Zika (ZIKV) viruses. Many critical discoveries within the RNA interference (RNAi) and genome editing revolutions were first pioneered in *Drosophila* because of its experimental advantages, and this progress is now permeating to the mosquito field. Genetic mechanisms and genomics studies of transposons established in *Drosophila* [1,2,3] are beginning to add to the extensive virology literature in mosquitoes [4,5].

Several transposons are characterized as “endogenous retroviruses” (ERVs), indicating the relation between transposons and viruses as invaders of animal cells that share many of the same genetic elements as viruses [6,7]. The distinctions between these invaders are that transposons are vertically transmitted from parent to offspring genomes and replicating themselves to new genomic locations, whereas viruses replicate in the cytoplasm or nucleus and transmit horizontally as budded particles between distinct individuals. We refer the reader to several reviews covering the rich biology of transposons and their relationship to viruses [8,9,10,11].

Animals have evolved defense mechanisms under immense evolutionary pressure to repress transposons and viruses in order to survive and reproduce as species, whereas the invaders need to evade these host defense mechanisms. This molecular “arms race” between the invading viruses and transposons and the animal host defenses is constantly ongoing at a heightened evolutionary pace to promote lineage specific genetic innovations [12,13,14]. RNAi is critically one of these host defense mechanisms conserved throughout nearly all branches of life to defend against transposons and viruses, and the general biochemical features of RNAi are also extensively covered in these other reviews [15,16,17,18].

The specific purpose of this review is to summarize, compare, and contrast how RNAi pathways have evolved rapidly between Drosophilids and Culicids to defend against transposons and viruses. In this review we raise some key questions (Box 1), we primarily discuss the Piwi protein and Piwi-interacting RNA (piRNA) pathways in these two Dipteran lineages and we touch upon the endogenous small interfering RNA (endo-siRNA), a canonical branch of RNAi. Although the microRNA (miRNA) branch of the RNAi pathway critically regulates gene expression during development for all animals [19,20], the role of microRNAs in transposon or virus suppression in insects has been more limited to a few studies [21,22,23], or has courted controversy [24,25,26]. We refer to [27] for further reading on this topic.

Box 1Key open questions regarding the Piwi/piRNA pathway differences between Drosophilids fruit flies and Culicids mosquitoes.What additional transposons’, viruses’, and genes’ silencing roles might explain the expansion of Piwi genes in culicine mosquitoes but not in anopheline mosquitoes and Drosophilids?Can mosquito Piwi pathways cause transcriptional chromatin silencing like in Drosophilids?What drives the immense diversity of piRNAs between mosquito species and strains, different transposons in their piRNA cluster loci or different states of virus infections?Can we harness or manipulate the mosquito Piwi/piRNA pathways to develop novel antiviral effectors for vector control strategies?

Other reviews have extensively discussed the RNAi/endo-siRNA [28,29,30,31,32] and Piwi/piRNA pathways [33,34,35,36,37] in animals, but here we will highlight the core facet of the two parallel sets of processing factors and effector Argonaute-family proteins defining the two classes of small RNAs in insects (Figure 1A). Whereas the miRNAs are processed by the RNAse III enzyme DICER1 and loaded into AGO1, endo-siRNAs are often processed by a second RNAse III enzyme DICER2 and loaded into AGO2 [38,39,40]. A much more complicated set of RNA export factors and nucleases completely distinct from DICERs process piRNAs that are then loaded into multiple Piwi proteins, such as PIWI, AUBERGINE, and AGO3 in *Drosophila* [34,41,42,43,44].

Although animal small RNA classes are technically defined/annotated by criteria different from originating locus and protein they are bound to [45], read length and sequence matching to existing databases can serve as an approximate proxy of the proportions of these small RNA classes (Figure 1B). For example, we observed varying proportions of miRNAs, endo-siRNAs, and piRNAs that form distinct profiles between Drosophilids and Culicidae [46]. These distinct small RNA class profiles may reflect the specific genomic, physiological, and evolutionary divergence between Drosophilids and Culicidae. Alternatively, different tissue expression patterns and unique geography of animals with different arbovirus infection statuses may also cause distinct small RNA profiles.

## 2. Genomic, Physiological and Evolutionary Differences between Drosophilids and Mosquitoes

To understand the biological context of the different small RNA class profiles between these two Dipterans, we first briefly review the physiological differences and similarities between these two insect clades. Fruit flies and mosquitoes share four similar life cycle stages, beginning with females laying fertilized eggs that hatch as larvae, the larvae undergo four instar stages and then enter a pupal stage to undergo metamorphosis, then eclose as adults (Figure 1C). Whereas both clades are cosmopolitan insects, their ecological niches and feeding physiologies are completely distinct. Fruit fly females can reproduce and the larvae thrive on the nutrients of rotting fruit, whereas female adult mosquitoes require a blood meal to develop her eggs which are oviposited in standing water for the larvae to hatch and develop in this completely aqueous environment.

Although ~260 million years (MY) of evolution has separated the *Culicidae* from *Drosophila* with distinct physiology and ecology (Figure 1D), the diversity within Drosophilids is narrower than Culicids, with most Drosophilids being related to each other within ~70 MY at the whole genome comparison scale that may also mirror the clade’s more narrow ecological niches [49,50]. In contrast, the two most prolific and biomedically important *Aedes* species, *Aedes aegypti* (*Ae. aegypti*) and *Aedes albopictus* (*Ae. albopictus*) are diverged by ~70 MY of evolution at the whole genome scale, followed by another ~100 MY separating the *Culex* lineage (Figure 1D) [51,52]. Finally, mosquitoes can be further divided into two major subclades that have ~48 MY of evolutionary separation: the culicine subclade that encompasses *Culex* and *Aedes* species, and the anophiline subclade with *Anopheles gambiae* (*An. gambiae*) as the representative species because of its widespread proliferation in tropical climates and the primary vector of malaria.

The subclade distinctions between anopheline and culicine mosquitoes not only reflect broader geographical preferences but also reflects two prominent biological and genomic characteristics. Only a small handful of arboviruses have been identified to be vectored by anopheline mosquitoes, ONNV and Mayaro virus [53,54], whereas culicine mosquitoes are infamous for their prolific capacity to vector EEEV and WNV in *Culex* strains and Chinkungunya virus, Yellow Fever virus, Zika virus, and dengue virus in *Aedes* strains [55,56,57,58,59]. Coincidently, the genome sizes and repetitive element contents of culicine mosquitoes is a magnitude larger than an anopheline mosquito genome and Drosophilid genomes (Figure 1E). In fact, Drosophilid and anopheline genomes are more related to each other through their compactness and minor fraction (<20%) of repetitive elements as well as generally limited proliferation of insect-specific viruses amongst Drosophilids [60].

## 3. Genome Transposon Composition and Germline Specificity of piRNAs in Drosophilids and Mosquitoes

Before we compare the endogenous piRNA clusters between Drosophilids and Culicids, we first review how genomic transposon content varies between Drosophilid, anopheline, and culicine lineages (Figure 1E). With its well-curated compact genome and ~140 manually annotated transposon families [61,62], *Drosophila* is the quintessential basis for dipteran comparisons. Despite some variations in estimates of transposon content across Drosophilid genomes [62,63,64], the relative abundance of transposons is conserved [62,65] and piRNA abundance correlates well with transposon load [66,67].

In contrast to fruit flies, culicine mosquitoes exhibit more drastic differences including larger genomes and extensive numbers of transposon families (Figure 1E) [68,69,70,71]. Interestingly, the transposon fractional proportion is similar between *An. gambiae* and Drosophilids genomes, perhaps also reflecting their similarly compact genome sizes. Among the large genome sizes of culicine mosquitoes [70,72], *Ae. albopictus* outranks *Culex quinquefasciatus* (*Cu. quinquefasciatus*) and *Ae. aegypti* having the greatest proportion of transposons relative to its genome (Figure 1E) [71,73]. Thus, mosquitoes represent a unique opportunity to investigate the relationship between expanding genomes, transposons, and their diverse defenses among closely and distantly related Dipteran species.

Among the many diverse defenses against transposons in insects, the piRNA pathway is a widely conserved mechanism to suppress the transposons which pose a perpetual threat to genome integrity [43,74,75,76]. Both Drosophilids and mosquitoes have the capacity to express piRNAs, but the pathway is best understood in *Drosophila melanogaster* (*D. melanogaster*). To discuss how Drosophilids piRNA pathways differ from mosquitoes, we highlight *Drosophila*-specific features. For example, the majority of *Drosophila* transposon-derived piRNAs are produced in the female germline [77] with a minor specialized somatic piRNA pathway in the follicle cells of the ovary [78] and in the fat body [79] (Figure 2A). In females, somatic piRNA cluster expression is lower than those present in the germline [80] (Figure 1B).

In contrast, male flies generally have reduced expression of piRNAs overall [77] and contain male-specific piRNA mapping patterns [81]. Such sex-specific differences suggest that multiple pathways for piRNA biogenesis may exist in fruit fly testes despite the limited role of piRNA-mediated silencing of transposons in the male [82]. The reasons for these differences in piRNAs and transposons between sexes remains unknown, but perhaps transposon silencing in the *Drosophila* male germline occurs in a narrow window because the *Drosophila* testes has fewer actively transcribing spermatogonia cells compared to the ovary where numerous large nurse cells and smaller follicle cells are actively dividing in the *Drosophila* egg chamber [83,84]. Nevertheless, there is still progress to further dissect Piwi pathways in Drosophilid spermatogonia such as possible nuclear to cytoplasmic shuttling of PIWI during spermatogonia mitosis [85]. Overall though, there are much fewer studies discussing piRNA biogenesis in the *Drosophila* male germline [86,87,88,89,90] compared to the female germline, which we predict may also be the trend in mosquitoes if spermatogenesis development patterns are similar.

The conventional wisdom is that transposon genomic load would positively correlate with greater expression of transposon-mapping piRNAs. However, culicine mosquitoes seem to challenge this notion considerably. For example, although >65% of the *Ae. aegypti* genome is filled with transposons, a low proportion (<5%) of piRNAs derived from these transposons [91]. In *Ae. albopictus*, the amount of transposon-mapping piRNAs are even less than *Ae. aegypti* despite an even larger genome with a greater fraction of repeats [92]. Interestingly, the female somatic tissues of *Cu. quinquefasciatus* have higher expression of transposon-mapping piRNAs, a feature not present in fruit flies [46].

However, the transposon-mapping piRNAs in anopheline mosquitoes mirror *Drosophila*: transposons are just ~15% genome size yet >70% of detected piRNAs are produced from transposon-containing piRNA clusters [75]. Perhaps, the lack of transposon-derived piRNAs in culicine mosquitoes are permitting transposons to replicate in and result in the expansion of these genomes [93,94]. With anopheline features looking more like *Drosophila* than culicine features, this raises the question as to what evolutionary event sparked this stark difference in genome transposon loads and transposon-mapping piRNAs in the culicine mosquito lineage. From these genome transposon loads, we also speculate that the anopheline mosquitoes are more closely related to the last common mosquito ancestor than the culcine mosquitoes.

Perhaps the tremendous genomic expansion in the culicine mosquitoes and greater capacity to be more prolific viral vectors compared to anopheline mosquitoes causes a biological need for culicine mosquitoes to express piRNAs in all tissues. In stark contrast to anopheline and Drosophilids, the piRNAs in culicine mosquitoes are detected in both germline and somatic cells and have similar expression levels to miRNAs (Figure 3A) [46,95]. This unique characteristic of bountiful somatic piRNAs in culicine mosquitoes prompts the question of why piRNAs are so restricted to the germline in *Drosophila*? Although some *Drosophila* piRNAs have been suggested to regulate embryonic development by degrading maternally deposited transcripts in the zygote which has also now been observed in *Ae. aegypti* [96,97,98,99], more of the *Ae. aegypti* piRNAs have been mapped to protein-coding genes abundantly expressed in the soma [100,101].

Ultimately, the transcriptional control of piRNA expression may be exceptionally narrow in *Drosophila*, as only four out of many tens of known *Drosophila* cell lines display robust piRNA expression (FGS, OSS/OSC, WRR1, Kc167) [102,103,104,105,106]. In contrast, every mosquito cell line we have analyzed expresses robust piRNAs [46] and many other lepidopteran insect cells also express abundant piRNAs [107,108,109,110]. In fact, the physiologically broader expression of piRNAs in mosquito somatic cells is a common trend in most other insect species [47].

## 4. The piRNA Clusters of *Drosophila* and Culicidae—Genomic Memory Bank of Transposon Invasion Versus Unique piRNA Clusters Containing Satellite DNA Repeats

*Drosophila* has led the small RNA/transposon field with the original definition of piRNAs clusters as master control loci of transposons [111] by extending the terminology first coined by Nobel laureate Barbara McClintock [112,113,114]. *Drosophila* possess two related but distinct piRNA pathways in the ovary, a somatic component in the follicle cells and a germline component in the nurse cells [78,115,116,117,118,119,120], where different piRNA cluster loci are uniquely regulated, yet both have their distinct transposon remnants. In the nurse cells, the master control locus termed *42AB* cluster is a dual-stranded piRNA cluster that produces piRNAs from both DNA strands to target transposons in the germline [121] (Figure 2B). This cluster depends on a specialized set of proteins known as the rhino, deadlock, and cutoff (RDC) complex to process various piRNAs with sequence homology to various nested transposons [67,122,123] (Figure 2D). Mutations in RDC components lead to major increases in various transposons, like *HetA* and *Burdock* [122,124,125,126,127,128,129].

In contrast to the *42AB* cluster expressed in nurse cells, the *flamenco* cluster [78] is only active in the follicle cells to generate a uni-strand piRNA cluster from an alternatively spliced transcript [130]. Before the field knew the *flamenco* locus was a piRNA cluster, it garnered its interesting name from being a regulator of the *gypsy* family of retrotransposons, which invade the germline exclusively through the somatic niche [131,132]. The constant repression of transposons in the germline likely results in the evolution of transposons (e.g., *gypsy*) that utilize enveloped viral particles to enter the germline [133,134]. Mutations in *flamenco* result in the loss of normal ovarian structure (via aberrant escort cells) due to the de-repression of transposons [135].

Like *Drosophila*, mosquito genomes also contain piRNA clusters (Figure 3B(i),B(ii)). However, mosquito piRNA clusters are particularly poorly understood compared to *Drosophila*. Earlier small RNA sequencing studies did first confirm the presence of piRNA clusters in *Ae. aegypti* [91], *An. gambiae* [136] and in *Ae. albopictus* [92], but the incomplete genomic assemblies and transposon annotations in some of these species hampered efforts for a comprehensive catalog of piRNA clusters. An individual species examination in *Ae. aegypti* discussed a mosquito piRNA cluster analogous to the *Drosophila flamenco* cluster [137], but we believe greater insight into mosquito piRNA cluster evolution is gained through multiple-species comparisons that include piRNAs from *Cu. quinquefasciatus* [46,96].

For example, cross-species piRNA cluster comparisons lead to the discovery of a conserved piRNA Cluster Locus (piRCL) composed of satellite DNA repeats in culicine mosquitoes [46,96]. Among the major anopheline and culicine mosquitoes, this piRCL is found near two genes of unknown function (Figure 3B(iii)) and did not appear to be restricted to any tissue or cell type [46]. Although these genes remain unknown, they serve as critical markers of genomic synteny enabling us to trace this piRCL’s evolution across ~170 MY of mosquito evolution. Interestingly, this piRCL is conserved in *An. gambiae* but only as a compact piRNA cluster without tandem repeats that generates abundant piRNAs from the associated gene’s 3′ UTR [46]. The anopheline piRCL may represent the ancestral mosquito compact locus that eventually expanded in culicine mosquitoes to include satellite DNA repeats in response to a new selective pressure on growing its genome size.

In terms of functionality, evidence suggests that this conserved piRNA cluster may be important for the maternal mRNA degradation pathway in mosquito zygotes and may be crucial for embryonic development in *Ae. aegypti* [96]. However, there is also evidence that this cluster is not exclusive to the early embryo and is found to be expressed in other tissues [46]. This finding puts forth the exciting possibility of this conserved piRCL having broader effects in other mosquito molecular processes (e.g., development and maintenance of germline). We also wonder if the innovation of culcine mosquitoes piRCLs composed mainly of satellite DNA repeats is linked to their capacity for expansive and repetitive genomes.

## 5. Rapid Evolution of Piwi Genes and piRNA Pathways

In response to transposons invading the soma and germline, Drosophilids have evolved distinct piRNA biogenesis mechanisms in each cell type. For example, somatically expressed single-strand clusters use canonical RNA Polymerase II transcription machinery to actively transcribe piRNA cluster transcripts [138,139]. In contrast, dual-strand clusters like *42AB* utilize other special proteins (RDC complex) in the female ovaries to promote piRNA cluster transcription (Figure 2D left) [122,123,124,125,128,140]. How and why the extraordinary divergence of piRNA biogenesis mechanisms arose still remains a mystery. Studies dissecting the complex protein interactions of the RDC complex suggest the role of adaptive evolution in shaping these distinct mechanisms [141,142].

piRNA-mediated silencing is predominantly associated with post-transcriptional silencing in the cytoplasm [30,31,32,33,34,35], however a single member of *D. melanogaster* PIWI does localize in the nucleus to direct transcriptional silencing through heterochromatin marks [143,144,145,146]. An extraordinarily complex relationship exists between the piRNA pathway and conserved heterochromatin-forming silencing complexes in *Drosophila* that continues to unveil new layers of regulation [145,146,147,148,149,150] (Figure 2D right). For example, very recent studies further uncovered the co-option of Mi-2, Rpd3, the Ccr4-Not complex, and the SUMO pathway with nuclear piRNA-guided transcriptional silencing [146,151,152]. Whether mosquito piRNA pathways can instigate chromatin silencing or utilize conserved heterochromatin formation machinery is still unknown.

Also unknown is whether mosquitoes contain distinct mosquito-specific piRNA biogenesis factors and silencing pathways for different cell types. However, we can suggest some insight from the evolution of Piwi gene members in mosquitoes. For instance, culicine mosquitoes showed an incredible expansion of the Piwi genes (up to seven orthologs) while anophelines retained a single *Drosophila* ortholog [46,153] (Figure 3D). Interestingly, both anophelines and culicine mosquitoes lack specific piRNA pathway genes (including *rhi*, *del*, and *cuff* among others) that are essential for dual-stranded cluster biogenesis in *Drosophila* [46,124,147,149,150]. Does this illustrate the lack of dual-stranded clusters in mosquitoes or hint to the possibility of a future discovery of unknown mosquito-specific proteins for piRNA biogenesis?

How the piRNA pathway evolved so rapidly between culicine and anopheline mosquitoes is a conundrum in the field. The presence of multiple Piwi genes may hint at selection developing protein/gene diversity against the evolution of transposons and viruses [154,155]. This is analogous to the expansion of APOBEC and KRAB-ZFN genes thought to be involved in arms races between eukaryotes and infections by retroviruses or retrotransposons [156,157,158,159]. Additionally, Piwi gene expansions could be a subfunctionalization response to the increased genomic complexity in culicine mosquitoes [46,96]. Perhaps during the genome expansion of Culicidae, Piwi genes were also expanded during the process and likely evolved culicids-specific mechanisms for piRNA biogenesis and silencing. Whatever the case, the evolution of culicine piRNA pathways have implications for transposon and virus defense of these efficient vectors.

## 6. Culicine Mosquitoes Display Extraordinarily Phased piRNA Biogenesis Patterns

Several other reviews have comprehensively covered the complex biogenesis mechanism of piRNAs that are clearly distinct from miRNA and siRNA biogenesis [34,43,118,160,161,162]. We summarize here the two key facets of piRNA biogenesis mechanisms that were first determined with *Drosophila* mutants and deep sequencing small RNA profiling [29,35,163] in order to then highlight one extraordinary aspect of mosquito piRNA biogenesis [46].

One facet of piRNA biogenesis is the ping-pong mechanism where two separate Piwi proteins can bind either the top strand or bottom strand piRNA that are off-set in sequence complementarity by 10 bases, whereby the endonucleolytic “slicing” activity of one Piwi protein helps define the 5′ base of the piRNA bound by the complementary Piwi protein [164]. Like *Drosophila*, mosquitoes have an AGO3 Piwi protein that likely performs ping-pong with the PIWI and AUB orthologs in *Drosophila* [92,155,165,166]. However, other than the Tudor protein *veneno* [167], the other ping-pong partner definitions amongst the culicine mosquito 6 different PIWI orthologs are not yet well defined.

A second piRNA biogenesis facet is sequential, phased generation of piRNAs in a head-to-tail fashion via additional endonucleases such as *zucchini*, *trimmer*, and *nibbler* [109,164,168,169,170] and helicases like *vasa* and *armitage* [171,172,173]. This phased generation of piRNAs may be kicked off by an initial ping-pong event via a “Trigger” piRNA that then leads to subsequent production of “Responder” piRNAs. This phasing pattern is discerned by various signal processing algorithms that reflect the initial Responder piRNA closest to the Trigger piRNA as the most abundant signal and then rapid decay and dissolution of the phasing pattern beyond the next couple of trailing piRNAs [164,168,174].

This phasing mechanism is deeply conserved across the animal kingdom, as shown by [164]. Notably, this study showed in the *Ae. aegypti* ovary that the 5′-to-5′ piRNA phasing profile is smoothly periodic and extends ~>4 piRNAs downstream of the initial peak. Most other animal’s piRNA phasing patterns are not as striking as this *Ae. aegypti* pattern. In fact, this remarkably periodic phasing pattern was only seen in two other culicine mosquitoes, *Ae. albopictus* and *Cu. quinquefasciatus* but absent in *An. gambiae* and Drosophilids [46] (Figure 3C). We do not yet understand how and why the 5′-to-5′ piRNA phasing profile is so striking in culicine mosquitoes, but we note that this pattern is maintained in the mosquito cell cultures that may be amenable to functional genomics and biochemical studies of this feature.

## 7. Viral piRNAs in *Drosophila* versus Mosquitoes: Minor in the Former and Major in the Latter?

As described before, *Drosophila* piRNAs’ main function is to silence transposons which can damage the germline genome [34,43,118,161,175]. However, a few studies suggests an antiviral immunity role for *Drosophila* viral piRNAs seen in the OSS cell line from mainly the plus strand of *Drosophila C virus* (DCV) and *American Nodavirus* (ANV), whereas other viruses like Noravirus, *Drosophila B virus*, and *Drosophila X virus* were generating viral siRNAs from both strands of the virus [176,177,178]. Although detection of viral piRNAs in intact *Drosophila* remains scant, two studies demonstrated increased replication of WNV and DXV replication after infecting *piwi and spn-E* mutants [177,178]. These studies are contrasted by Petit et al. who argue that the Piwi pathway is not required for antiviral responses in *Drosophila* where no SINV piRNA were detected because the AGO2/endo-siRNA pathways may provide the bulk of the antiviral responses [179].

Although the AGO2/endo-siRNAs also appear to silence transposons [29,102,180,181,182,183] and provide antiviral immunity against viruses [177,178,184,185,186,187,188,189,190,191,192,193], the interplay between antiviral and transposon silencing pathways remains elusive. However, a recent study in *D. melanogaster* and *D. simulans* demonstrated a significant decrease of transposon activity exclusively in the soma of SINV-infected flies [194]. In the same study, an impaired siRNA pathway led to the increased production of viral piRNAs, a possibly oversaturated piRNA machinery, and an increase in transposon transcripts, suggesting a tight synergy between both pathways [194]. An interesting implication is a complex genetic cross talk between the endo-siRNA antiviral response and transposon regulation by piRNAs in *Drosophila*, with an open question of whether a somatic role for *piwi* and *spn-E* and some Piwi components are underlying this genetic cross talk.

A fascinating aspect of mosquito vector biology is their ability to allow systemic and persistent viral infections with high viral loads. In mosquitoes, RNAi is widely known to be the predominant innate immune program against viruses [195]. However, the extent to which some or all of these small RNA pathways limit viral propagation is unknown. Recently, studies have shown the ability of both mosquitoes and cell lines to give rise to virus-derived viral piRNAs (vpiRNAs) upon virus infection [46,76,153,155,165,167,196,197,198,199]. Unlike mosquitoes, Drosophilids appear to lack viral piRNAs in their soma and germ cells despite viral infections (Figure 1B) [179]. This difference among Dipterans can be attributed to the capacity of mosquitoes to be prolific flavivirus vectors.

Indeed, isolated mosquito cell lines also maintain persistent viral infections despite being cultured away from the animal for decades. The ability to maintain low viral titers in specific infected-mosquito cell lines illustrates the effectiveness of the antiviral immune program [200,201,202,203]. Although the *Ae. albopictus* C6/36 cell line is unable to mount antiviral siRNAs due to a *dcr-2* deficiency [165,198], these cells are extremely useful for the propagation of viruses such as ZIKV [204,205]. To overcome the RNAi deficiency, C6/36 cells must be supplemented with synthetic siRNAs to reduce viral infection [206].

An interesting facet of vpiRNA production in culicine mosquitoes is how quickly they can produce vpiRNAs. Depending on the virus and medium (cell culture or mosquitoes), there is evidence suggesting piRNAs can be detected within a few days to 1 week [46,196,207,208]. For example, production of siRNAs in mosquito cells were prominent during the early stages of Rift Valley fever virus infection, whereas vpiRNAs significantly accumulated later on [208]. Similarly, in CHIKV-infected *Ae. albopictus* mosquitoes, vpiRNAs were detected in low levels at three days but became abundant after nine days [196]. Likewise, vpiRNAs in ZIKV-infected mosquitoes were detected within one week [46,207]. The differing results from these studies likely reflect the complexity of virus-mosquito interactions in vivo. More importantly, these results point to an underlying partnership between the siRNA and piRNA pathways for antiviral immunity in mosquitoes.

Viral DNA fragments have been found to be integrated into host genomes [8,209,210]. Whereas viral integrations are prevalent in Culicidae genomes, amazingly, *Aedes* mosquitoes have 10 times more EVEs than other mosquito species [211,212,213,214]. This striking capacity may reflect on the repetitive genomes of *Ae. aegypti* and *Ae. albopictus*. The highly repetitive genome of *Ae. albopictus* ranks higher in more EVEs (located outside piRNA clusters) compared to *Ae. aegypti* [215]. Despite being more similar in size and repeat content to *D. melanogaster*, *An. gambiae* genomes also have EVEs inside piRNA clusters [212,215], likely reflecting mosquito-specific evolution. The reason for these species-specific differences between having EVEs outside or inside piRNA clusters still remains a mystery. Further characterizations of EVEs across species will be required to further understand this evolutionary outcome.

Recently, a unique piRNA cluster was discovered in *Ae. aegypti* mosquitoes that resembled *D. melanogaster*’s *flamenco* locus [137]. This 143 kb “*flamenco*-like locus” is a single unidirectional cluster composed of several non-retroviral EVEs and transposons with the capacity to express both EVE- and transposon-piRNAs. The lack of EVEs in *Drosophila* pose interesting questions about how EVEs eventually evolved to be maintained in this *flamenco*-like locus in *Ae. aegypti*. Whether this locus is conserved in other culicine species remains unknown. Furthermore, it is unclear if this locus plays a crucial role for antiviral immunity in mosquitoes. Perhaps it depends on viruses that cause persistent viral infections. Recent evidence suggests some non-retroviral EVEs in *Ae. aegypti* mosquitoes can provide antiviral immunity against cell fusing agent virus (CFAV), a widely circulating mosquito virus [216]. An interesting question is how highly repetitive genomes (e.g., culicine mosquitoes) maintain antiviral EVEs located in either inside or outside piRNA clusters.

## 8. Is the Role of Active Transposons Causing Hybrid Fertility in *Drosophila* Conserved in Mosquitoes?

In a classic genetic phenomenon of hybrid sterility, pioneering studies in *Drosophila* have made important associations of this phenotype to transposon control by piRNAs [217,218,219,220]. The *P-element* transposon is one prominent agent driving the molecular basis of this phenomenon [2,221,222,223,224,225,226]. Female sterility occurs in F1 progeny from crosses between a *P-element*-containing male and a female lacking *P-elements*. In the reciprocal cross where the female possesses the *P-elements*, the F1 hybrids are fertile (due to maternal contribution of piRNAs [227]). The dysgenic cross where F1 females are sterile because the lack of *P-element* silencing is thought to cause defects like chromosomal breakage, germline cell apoptosis, and an increase in point mutations that harm female germ cell development [228].

The study of how *Drosophila* regulates *P-element* control is still an active and important topic. For example, *P-elements* have completely invaded nearly all wild *Drosophila* populations, but interplay between how quickly the piRNA pathway adapts to controlling *P-element* silencing is still being investigated [229,230,231]. An outstanding question is determining to what extent *P-element* copy number affects the severity of hybrid dysgenesis. Both recent and early studies demonstrated a positive correlation between hybrid dysgenesis severity and *P-element* copy number [2,3,225,232], however, others report a weak [233] or lack of correlation [234,235,236]. The discrepancy may be due to species-specific differences, intraspecies genetic differences, *P-element* structural variation, or piRNA cluster size [234,237,238,239].

Could fertility studies in mosquito hybrids gain insight from *Drosophila P-element* hybrid dysgenesis studies? We are still at the early stages of determining the underlying genetic mechanisms behind how some crosses are sterile whereas other crosses from the same set of species but of different geographical regions can become fertile. For example, in Hawaii, nonviable eggs were produced in mating between *Ae. albopictus* females and *Ae. aegypti* males, but in reciprocal mating, hybrids were produced [240]. However, when similar crosses were performed on Florida strains, no hybrids were produced [241]. Could it be possible that a mosquito’s unique repertoire of transposon-derived or viral-derived piRNAs contribute to this phenomenon? A survey of mosquito strains around the world may give insights into this question.

We propose that studying piRNA pathways and transposon regulation factors could lead to new insights in earlier studies of hybrid sterility in mosquitoes. Examples of hybrid sterility studies are from *An. gambiae* sterile hybrids [242] that suggest an unidentified incompatibility factors on the X chromosome [243,244,245], and meiotic abnormalities from the unpairing of sex chromosomes [243]. In addition, early work in *Anopheles arabiensis* hybrids discovered a transposon named *Odysseus* present at the junction of an inversion, with distinct distributions of this element in different strains [246]. The role of inversions in adaptation and speciation in the *An. gambiae* complex have been extensively studied and reviewed in [247]. Could there be a link between anopheline hybrid incompatibilities and transposons, with the possibility of piRNAs modulating this phenomenon? We also speculate if hybrid sterility in mosquitoes could be linked to the different compositions of piRNAs coming from virus and endogenous viral elements (EVEs).

## 9. How Conserved is the AGO2 Endo-siRNA Pathway between *Drosophila* and Mosquitoes?

In *Drosophila*, endogenous siRNAs (endo-siRNAs) were first defined as being a distinct class from the ‘repeat-associated siRNAs’ [248,249,250,251,252] that were later classified as piRNAs [67,120,253]. Although one study described endo-siRNAs in both the germline and soma tissues with no unique differences in biogenesis [254], the main distinction of endo-siRNAs from piRNAs in *Drosophila* is a broader role for siRNAs in the soma toward antiviral defense [186], transposon defense [255], and gene regulation [256] (Figure 2A). In contrast to exogenous siRNAs, endo-siRNAs are derived from the genome itself rather than an exogenous source such as viruses and use distinct biogenesis pathways [257]. With the slicing role of AGO2 being crucial for all siRNA pathways [182,186,255,258,259], fruit flies utilize AGO2 for both exo- and endo-siRNA pathways, suggesting co-evolution of two distinct small RNA pathways [31].

In mosquitoes, little is known about the endo-siRNA pathway and few studies document the presence of siRNAs. The role of endo-siRNAs in antiviral immunity [260] and transposon defense in the gonads [261] demonstrates the versatility of this pathway in defense mechanisms. Interestingly, the majority of mosquito siRNA studies point to the exo-siRNA pathway as a major process against viral infection [200,262,263,264,265,266,267]. Why the exo-siRNA pathway has a dominant role in antiviral immunity is unknown. Insight into the host-virus evolution may explain why the endo-siRNA pathway has a smaller role against viruses. For example, the mosquito exo-siRNA pathway genes appear to undergo rapid, positive, and diversifying selection [268,269].

Since mosquitoes are prominent vectors for viruses compared to Drosophilids, it is likely that an evolutionary arms race between mosquitoes and viruses is compelled to diversify the siRNA pathway against the virus or associated viral suppressors of RNAi. These viral counter-defenses drive adaptations in the host immune genes and can ultimately lead to highly specific host and virus interactions (e.g., the exo-siRNA pathway) [270]. Future comparative studies on the evolution of siRNA pathways between fruit flies and mosquitoes will provide valuable insights into the species-specific adaptations against viruses.

## 10. RNAi Pathways as Applications for Mosquito Vector Control

There are practical applications to understanding the piRNA pathway in mosquitoes such as the development of novel tools and approaches for mosquito vector control. For example, scientists leveraged their understanding of the miRNA pathway in mosquitoes to build synthetic miRNAs in transgenic *Ae. aegypti* to target RNA arboviruses such as ZIKV, CHIKV, and DENV [205,271]. Another report also developed transgenic *Ae. aegypti* mosquitoes that robustly expressed inverted-repeat sequences derived from the DENV-2 genomic RNA to trigger the RNAi response and confer significant resistance to this virus [272]. The development of these tools against flaviviruses are especially important for the field of vector control because traditional tools such as insecticides are increasingly becoming inefficient due to mosquitoes acquiring insecticide resistance [273].

Novel genetic tools are becoming a promising alternative to curb the spread of harmful mosquito-borne pathogens. A question still remains: can piRNAs be used for targeting RNA viruses in transgenic mosquitoes and be as efficient as similar strategies mentioned above? Rapid advances in gene editing of mosquitoes and mosquito cell cultures will be the key towards harnessing mosquito small RNA biological pathways so that one day we can tame and domesticate these diverse Dipteran defenses to serve humankind.

## 11. Final Thoughts: Strengthening the Fly-Mosquito Partnership for Future Biomedical Progress

Many fundamental molecular and developmental biology discoveries, including the Piwi pathway were first made in *Drosophila* [274,275,276] due in part to its simple husbandry, compact genome size and pioneering genetic tools. Mosquito researchers can be envious of *Drosophila*’s genetic tools, such as unparalleled annotation and manipulation of its genome [277], plethora of genetic markers and balancer chromosomes, and binary transgenes to trigger RNAi and CRISPR genome editing [278,279]. We envision that more and more investigators like in the Akbari and Lau labs [234,280,281,282,283] and other notable Drosophilists [284,285] will leverage their initial foundation of studies in *Drosophila* and extend their expertise to the mosquitoes.

Although mosquitoes have more complicated rearing protocols compared to *Drosophila* (e.g., larval development in water, adult rearing in cages, blood for production of eggs, humidity and temperature requirements), these organisms are rising as future models for studying insect development and molecular studies. With some advantages like *Aedes* embryos undergoing diapause as a useful storage trick in experiments [286], and being closely related to Drosophilids, we hope that future genomic comparisons and greater development of useful genetic tools in mosquitoes [287,288,289,290] will encourage more funding efforts and investigators supporting mosquito research.

Indeed, more work is required for both insects to illuminate the biology of small RNA pathways, but progress lags in mosquitoes because the field still needs better curated mosquito genomes. Currently, mosquitoes lack well annotated genomes and contain an astonishing number of repeats which ultimately makes genome annotation and sequence assembly more challenging. Some recent progress has been made to further improve *Ae. albopictu*s’ highly repetitive genome sequence [73] and to uncover its developmental transcriptome [291]. In addition, efforts to improve the *Ae. aegypti* [51] and *An. stephensi* [292] genomes were also accomplished.

The field would also benefit from a sampling of wild mosquito genomes from all over the world. The use of lab strains has advantages, but ultimately, wild mosquitoes are valuable resources to understand how mosquito genomes differ based on location and to understand if local exposure of particular pathogens can drive the evolution of small RNA pathways in distinct ways. For example, our study [46] detected differences in *Ae. aegypti* small RNA profiles and different loads of viral piRNAs between various lab strains versus other strains more recently isolated from the wild, but we do not know how variable are the underlying genomic piRCL sequences. In addition, recent studies in *Drosophila* showed ongoing transposon invasions and environmentally induced changes continuing to drive piRNA cluster evolution [139,293].

An open question is why are somatic piRNAs prolific in Culicidae but largely absent in Drosophilids. Is it possible to ectopically express somatic piRNAs in *Drosophila*? Are there specific *Drosophila* factors/genes that prohibit somatic piRNAs and limit fruit flies from efficiently vectoring viruses? Perhaps the capacity of horizontal gene transfer modes between Drosophilids may be an explanation. Fruit flies consume fruit and fungi while female mosquitoes feed on sugar and blood to obtain necessary nutrients for egg production. Mosquitoes acquire infection by obtaining viral pathogens from animal blood. Could blood feeding be a route for horizontal gene transfer if midgut digestion is not thoroughly complete? Further characterization of wild insect lines may shed some light on this matter.

There is a need to apply genetics and functional genomics in the mosquito to better understand the small RNA pathway features. New genetic tools for mosquitoes are still emerging, such as germline editing using CRISPR in mosquitoes [288,289] and novel Cas9 delivery via ReMOT [294,295,296] that may enable creating Piwi pathway mosquito mutants. These mutants could inform on whether infertility results from loss of mosquito Piwi genes, but the redundancy of Piwi genes in mosquitoes may also obscure an overt phenotype. Some groups are also leveraging CRISPR and RNAi in mosquito cell cultures for knock-out and knockdown studies of small RNA pathway genes to study viral infection responses [46,76,197,216,297,298,299], and we anticipate that future genome-wide intervention resources for mosquito will be developed when the mosquito genomic curations and annotations improve to the levels existing for *Drosophila*.

## Figures and Tables

**Figure 1 cells-09-02180-f001:**
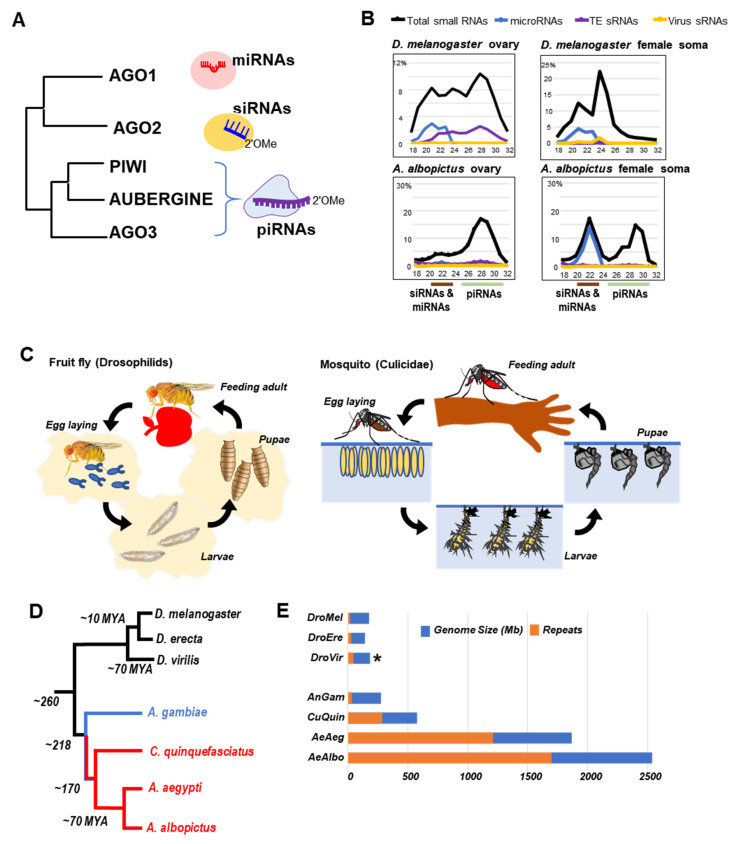
Comparing small RNA pathways between fruit flies (Drosophilids) versus mosquitoes (Culicids). (**A**) Phylogeny of the Argonaute-(Ago) family proteins from *Drosophila* with diagrams of these Ago-small RNA complexes. (**B**) Size distribution graphs representing the Drosophilid and Culicid small RNA profiles of female animals not directly infected with arboviruses. *D. melanogaster* data from [47], whereas *Ae. albopictus* data is from [46]. (**C**) Biological and niche differences and life-cycle commonalities. (**D**) Evolutionary phylogeny of Drosophilids versus Culicids. (**E**) Genome size and repetitive sequence composition differences based on current NCBI/Genbank genome assemblies and annotations of these Dipteran species. The asterisk notes that the current assemblies of the *D. virilis* genome are smaller than the actual genome because of sizeable satellite DNA absent from the reference assemblies [48].

**Figure 2 cells-09-02180-f002:**
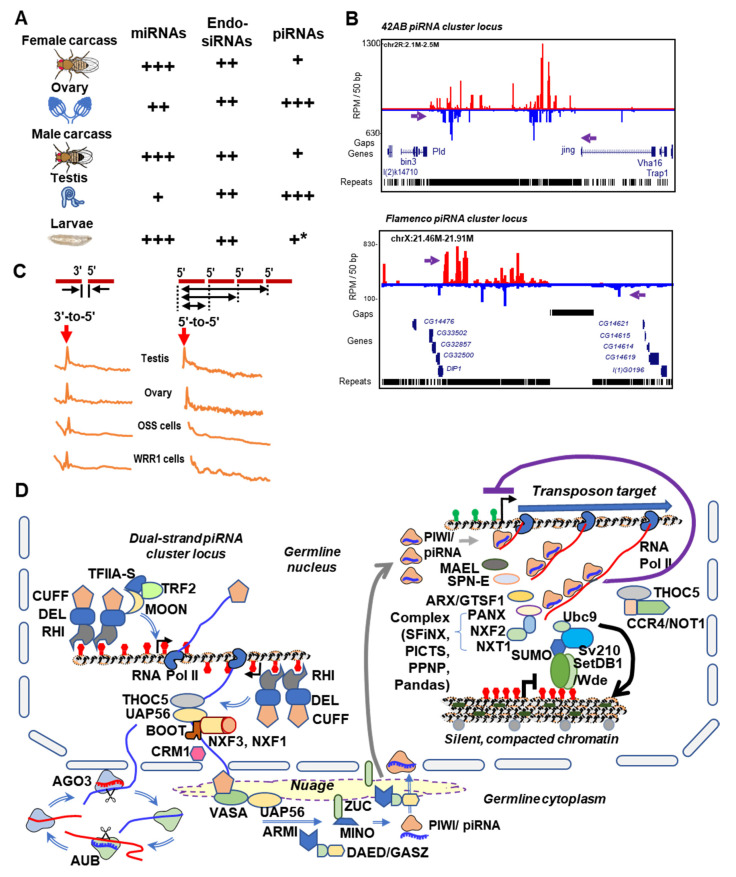
*Drosophila*-focused Piwi/piRNA pathway features. (**A**) *Drosophila* piRNA expression is restricted to germline, with an asterisk for piRNAs in larvae libraries that include the larval gonads. (**B**) Configuration of two notable *Drosophila* piRNA cluster loci, *42AB* and *flamenco*. (**C**) *Drosophila* piRNA phasing patterns analyses with red arrows pointing to salient signature of piRNAs juxtaposed to each other from the precursor transcript. (**D**) Schematic of *Drosophila*-specific Piwi/piRNA pathway silencing complexes such as the Rhino-Deadlock-Cutoff complex that promotes piRNA biogenesis from the 42AB locus (left) and a transcriptional gene silencing network (right; various names like SFiNX, PICTS, PNPP, Pandas complex) and Sv210 and CCR4/NOT1 complex that target transposon silencing at the chromatin level in the nucleus.

**Figure 3 cells-09-02180-f003:**
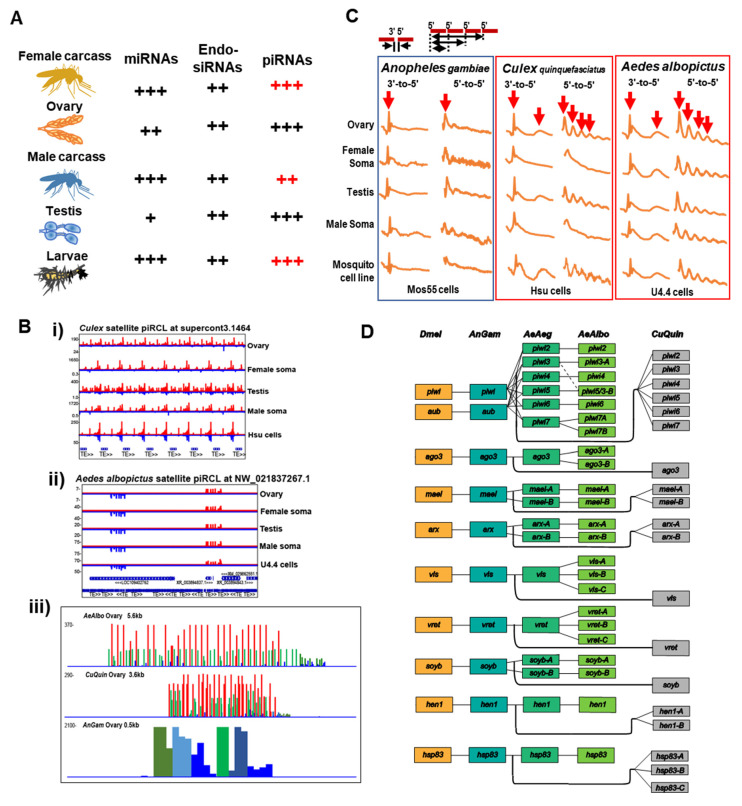
Mosquito-focused Piwi/piRNA pathway features. (**A**) Mosquito piRNA expression is notably broad and almost as ubiquitous as miRNAs in being expressed both in the soma and gonads. (**B**) Genome browser snapshots showing the novel configuration of piRNA cluster loci (piRCL) in culicine mosquitoes as satellite piRCLs, with (**i**) a *Culex* satellite piRCL, (**ii**) an *Aedes* satellite piRCL, and (**iii**) a deeply conserved piRCL that is a greatly expanded satellite repeat in culicine mosquitoes but is very compact in the anopheline species. (**C**) Mosquito piRNA phasing patterns analyses with red arrows pointing to salient signature of piRNAs juxtaposed to each other from the precursor transcript. The periodicity of piRNA phasing is most apparent in culicine mosquitoes. (**D**) Multiple Piwi pathway genes underwent homolog expansion amongst culicine mosquito lineages, most notably the expansion of PIWI homologs.

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
