# Peer review of "Diverse Defenses: A Perspective Comparing Dipteran Piwi-piRNA Pathways"

_cells, 2020, doi:10.3390/cells9102180_

Round 1

Reviewer 1 Report

The manuscript constitutes a comprehensive review of RNA interference (RNAi) pathways in two insect families, i.e. the Drosophilidae fruit flies and the Culicidae mosquitoes. On account of the recent novel knowledge regarding the role of RNAi in animal species, it is a very interesting topic of high importance. The review is well organised and well structured in a very reasonable manner. Also the paper is well written in an easy-to-understand language. I propose only two minor changes and one suggestion that I have, regardless of which I recommend publication. In general the authors have done a very good effort.

Minor changes

lines 38-39: Rephrase as follows: Transposons are characterised as “endogenous retroviruses” (ERVs), indicating the relation between Transposons and viruses as invaders of animal cells that share many of the same genetic elements as viruses

line 141: delete “with”

Suggestion

Since as also mentioned in the section "Final thoughts: strengthening the fly-mosquito partnership for future biomedical progress", RNAi pathways may be useful as tools for vector control, I only suggest to move this part and discuss it a little bit more in a separate section.

Author Response

lines 38-39: Rephrase as follows: Transposons are characterised as “endogenous retroviruses” (ERVs), indicating the relation between Transposons and viruses as invaders of animal cells that share many of the same genetic elements as viruses

Thank you for the suggested revision. We have revised the manuscript to reflect this change.

line 141: delete “with”

We have deleted this word.

Suggestion

Since as also mentioned in the section "Final thoughts: strengthening the fly-mosquito partnership for future biomedical progress", RNAi pathways may be useful as tools for vector control, I only suggest to move this part and discuss it a little bit more in a separate section.

We have moved this part as its own section located before the “Final thoughts” section. A little more information about RNAi as a vector control tool has been added.

Reviewer 2 Report

This is an extremely well written review. It flows well, is easy to read, and is very educational (specifically to an audience not familiar with PiWi/piRNA pathway). It would be ideal if the authors could include a "key questions" box that highlights some of the most important questions that could be addressed by the field. The authors have highlighted some in the summary, but a clearer set of questions will make this review exceptional.

Author Response

This is an extremely well written review. It flows well, is easy to read, and is very educational (specifically to an audience not familiar with PiWi/piRNA pathway). It would be ideal if the authors could include a "key questions" box that highlights some of the most important questions that could be addressed by the field. The authors have highlighted some in the summary, but a clearer set of questions will make this review exceptional.

We have added a “key questions” box at the end of the manuscript for outstanding questions in the field.

BOX 1.  Key open questions regarding the Piwi/piRNA pathway differences between Drosophilids fruit flies and Culicids mosquitoes.

  1. What additional transposons’, viruses’, and genes’ silencing roles might explain the expansion of Piwi genes in culicine mosquitoes but not in anopheline mosquitoes and Drosophilids?
  2. Can mosquito Piwi pathways cause transcriptional chromatin silencing like in Drosophilids?
  3. What drives the immense diversity of piRNAs between mosquito species and strains, different transposons in their piRNA cluster loci or different states of virus infections?
  4. Can we harness or manipulate the mosquito Piwi/piRNA pathways to develop novel antiviral effectors for vector control strategies?

Reviewer 3 Report

This is an interesting and well-cited perspective-style review paper comparing and contrasting piRNAs in drosophilids and mosquitos. I especially like the figures. The manuscript would be improved by substantial changes to the section and paragraph structure, but it is not necessary in my opinion for this to be a useful contribution to the literature.

I have only one major point, which relates to a potential misconception about EVEs. At the least, the language is unacceptably ambiguous:

  1. 449: “Have the capacity” suggests that this is a property mediated by the virus, e.g. enabled by virally-encoded enzymes. But for most of the EVEs in these citations, specifically the non-retroviral EVEs (which later statements make me think that the author is using EVEs to indicate, to the exclusion of retroviral EVEs e.g. by the statement “lack of EVEs in Drosophila” in line 463), this is not the case. Except for retroviruses (a minor component of the mosquito EVE-ome), I would argue that EVEs are not integrated in a manner akin to transposons. EVEs are “dead ends” for the involved viral nucleic acids, in terms of their “life history” as replicons with generation times shorter than that of their host. This is in distinct contrast to transposons, for which integration is fundamental to their replication as selfish genetic elements.

The rest of my comments are minor:

Line 32: recommend “intently” -> “intensely.”

  1. 33: awkward because some members of the list are diseases, some viruses.
  2. 38-39: this suggests ERVs are transposons but not viruses, but some meet the subsequently listed criteria e.g. horizontal transfer. It may be clarifying to note that definitions of transposons and viruses overlap in the case of some ERVs.
  3. 42: some infectious viruses replicate in the nucleus.
  4. 81: refers to an asterisk that I cannot find.
  5. 84: some reference would be useful to evidence this statement. miRNA are defined/annotated by criteria different from originating locus and protein they are bound to (Ambros V et al., RNA 2003).
  6. 104-111: I invite the authors to speculate here whether/how the vastly different repetitive component of the genomes carried by the members of the different families (Fig 1E) influence predictions of last common ancestor based on whole genome comparison, though I understand it is outside their major focus.
  7. 152-153 regarding somatic piRNAs seemingly contradicts line 157 “restricted to germline.” Could say “restricted to gonad,” but what then of the fat body (Jones BC et al., Nat Comm 2016)?
  8. 189-190: the argument is less clear than it could be. Would another way to say this would be that lack of transposon-derived piRNAs is permitting transposons to replicate in, and thus expand, culicine genomes unchecked?
  9. 191: consider “raises the question.”
  10. 195-197: please clarify. Does “reflect” mean ”causes?”
  11. 203-204: please clarify. How are the words “gene regulation” functioning in this sentence? Noun adjunct?
  12. 217-218: “patterns” is redundant. I suggest “such as” -> “as only” and remove “that.”
  13. 242: is “exclusively” positioned correctly? What about “which invade the germline exclusively through the somatic niche.”
  14. 244: “envelope particles” should either be “enveloped viral particles” or “envelope proteins.” Your perspective, your prerogative to use “clever” in this context. The Publication manual of the APA recommends avoiding anthropomorphism, not sure about the journal Cells.
  15. 258: unclear why you start a new paragraph here as the thought is closely linked to the previous sentence.
  16. 290: Several Piwi family members localize to the nucleus, so “a single member of Piwi” needs further qualification or to be removed. I suggest specifying D. mel PIWI to clarify why it is written in majuscule here.
  17. 302: Are there 7 orthologs total in the family but only 6 orthologs per species? This is not as clear as it could be, especially when 6 orthologs are mentioned e.g. line 334.
  18. 313: Cas family diversification in prokaryotes may not be the best analogy for naturally-selected protein duplications in eukaryotes. I recommend analogizing to duplications of for example APOBEC genes or KRAB-ZFN genes thought to be involved in arms races in other eukaryotes. Nor is the citation chosen a particularly compelling citation for the analogy chosen.
  19. 331: “Mosquitos like Drosophila” needs two commas to set off the appositive phrase. “Like Drosophila, mosquitos” would only need one.
  20. 356-358: this could be shorter.
  21. 354-395: This is a particularly interesting section and contains the interesting speculations that I consider valuable in a perspective piece. But it may be difficult for readers who are not already familiar with “piRNAs coming from viruses and EVEs” (line 395), which have not been introduced yet, to follow the logic. Couldn’t this entire section follow the section that starts at L. 397?
  22. 399: This is an awkward sentence. The potential antiviral role for piRNAs comes seemingly out of nowhere; why/how does this potential exist, or in what literature has this potential role been proposed? Also “silence” is rarely linked with the preposition “from.”  
  23. 406: “that argue” to “who argue.”
  24. 427: Did Fig. 1B come from virally-infected drosophilids?
  25. 454: “Very similar” -> “more similar in size and repeat content”
  26. 470: I don’t understand this sentence, particularly how the prepositional phrase “inside or outside piRNA clusters that ultimately lead to the antiviral response” is intended to modify “such EVEs.”
  27. 489-490: “but perhaps…” this part of the sentence is weak. The following paragraph should be included in this paragraph.
  28. 509: The tone here may not be what the senior authors are going for.
  29. Lines 509-510 introduce many self-citations of questionable value based on the context.

Author Response

  1. 449: “Have the capacity” suggests that this is a property mediated by the virus, e.g. enabled by virally-encoded enzymes. But for most of the EVEs in these citations, specifically the non-retroviral EVEs (which later statements make me think that the author is using EVEs to indicate, to the exclusion of retroviral EVEs e.g. by the statement “lack of EVEs in Drosophila” in line 463), this is not the case. Except for retroviruses (a minor component of the mosquito EVE-ome), I would argue that EVEs are not integrated in a manner akin to transposons. EVEs are “dead ends” for the involved viral nucleic acids, in terms of their “life history” as replicons with generation times shorter than that of their host. This is in distinct contrast to transposons, for which integration is fundamental to their replication as selfish genetic elements.

We thank the reviewer in noticing this error and we apologize for the incorrect statement. This statement has been changed in the manuscript to say the following: “Viral DNA fragments have been found to be integrated into host genomes”

The rest of my comments are minor:

Line 32: recommend “intently” -> “intensely.”

Changed.

33: awkward because some members of the list are diseases, some viruses.

We have modified the statement to only reflect the pathogens transmitted. The revised statement: “…serious pathogens that affect humans like Plasmodium falciparum (malaria parasite), West Nile virus (WNV), chikungunya virus (CHIKV), Yellow fever virus (YFV), dengue (DENV) and Zika (ZIKV) viruses”

38-39: this suggests ERVs are transposons but not viruses, but some meet the subsequently listed criteria e.g. horizontal transfer. It may be clarifying to note that definitions of transposons and viruses overlap in the case of some ERVs.

Another reviewer noticed this error as well and we have addressed it to say the following: “Transposons are characterized as “endogenous retroviruses” (ERVs), indicating the relation between transposons and viruses as invaders of animal cells that share many of the same genetic elements as viruses”

42: some infectious viruses replicate in the nucleus.

We have modified the text to reflect some viruses replicate in the nucleus.

81: refers to an asterisk that I cannot find.

Thanks for pointing out this error in the figure that was inserted into the Word Doc.  We have corrected this with a proper figure that includes the asterisk.

84: some reference would be useful to evidence this statement. miRNA are defined/annotated by criteria different from originating locus and protein they are bound to (Ambros V et al., RNA 2003).

Sentence modified and citation added as follows: “…defined/annotated by criteria different from originating locus and protein they are bound to (Ambros et al. 2003)…”

104-111: I invite the authors to speculate here whether/how the vastly different repetitive component of the genomes carried by the members of the different families (Fig 1E) influence predictions of last common ancestor based on whole genome comparison, though I understand it is outside their major focus.

We added the following sentence to the text: “From these genome transposon loads, we also speculate that the anopheline mosquitoes are more closely related to the last common mosquito ancestor than the culicine mosquitoes.”

152-153 regarding somatic piRNAs seemingly contradicts line 157 “restricted to germline.” Could say “restricted to gonad,” but what then of the fat body (Jones BC et al., Nat Comm 2016)?

Figure caption has been revised to remove the contradiction. Fat body piRNAs and citation are now included in sentence describing somatic piRNAs.

189-190: the argument is less clear than it could be. Would another way to say this would be that lack of transposon-derived piRNAs is permitting transposons to replicate in, and thus expand, culicine genomes unchecked?

We apologize for the unclear statement. The sentence has been modified to now read: “Perhaps, the lack of transposon-derived piRNAs in culicine mosquitoes are permitting transposons to replicate in and result in the expansion of these genomes”

191: consider “raises the question.”

Modified.

195-197: please clarify. Does “reflect” mean ”causes?”

Modified.

203-204: please clarify. How are the words “gene regulation” functioning in this sentence? Noun adjunct?

We have removed the phrase “gene regulation” from this sentence.

217-218: “patterns” is redundant. I suggest “such as” -> “as only” and remove “that.”

Sentence is now modified to reflect the suggestions: “Ultimately, the transcriptional control of piRNA expression may be exceptionally narrow in Drosophila, as only four out of many tens of known Drosophila cell lines display robust piRNA expression (FGS, OSS/OSC, WRR1, Kc167)”

242: is “exclusively” positioned correctly? What about “which invade the germline exclusively through the somatic niche.”

We changed the position of the word “exclusively” in this sentence.

244: “envelope particles” should either be “enveloped viral particles” or “envelope proteins.” Your perspective, your prerogative to use “clever” in this context. The Publication manual of the APA recommends avoiding anthropomorphism, not sure about the journal Cells.

We have removed the phrase “clever enough to…” and modified the sentence as follows: “The constant repression of transposons in the germline likely results in the evolution of transposons (e.g. gypsy) that utilize enveloped viral particles to enter the germline”

258: unclear why you start a new paragraph here as the thought is closely linked to the previous sentence.

We divided the two paragraphs to keep the text block from being too large to read.

290: Several Piwi family members localize to the nucleus, so “a single member of Piwi” needs further qualification or to be removed. I suggest specifying D. mel PIWI to clarify why it is written in majuscule here.

The sentence has been modified to read: “…however a single member of D. melanogaster PIWI, the prototype PIWI protein does localize in the nucleus…”

302: Are there 7 orthologs total in the family but only 6 orthologs per species? This is not as clear as it could be, especially when 6 orthologs are mentioned e.g. line 334.

Thank you for discovering this error. We have revised Figure 3D to properly show the 7 Piwi orthologs in Aedes albopictus.

313: Cas family diversification in prokaryotes may not be the best analogy for naturally-selected protein duplications in eukaryotes. I recommend analogizing to duplications of for example APOBEC genes or KRAB-ZFN genes thought to be involved in arms races in other eukaryotes. Nor is the citation chosen a particularly compelling citation for the analogy chosen.

Thank you for the suggestion. We have modified the text to include mentioning the expansion of APOBEC and KRAB-ZFN genes with appropriate citations.

331: “Mosquitos like Drosophila” needs two commas to set off the appositive phrase. “Like Drosophila, mosquitos” would only need one.

Modified.

356-358: this could be shorter.

We have shortened this section according to the reviewer’s recommendation.

354-395: This is a particularly interesting section and contains the interesting speculations that I consider valuable in a perspective piece. But it may be difficult for readers who are not already familiar with “piRNAs coming from viruses and EVEs” (line 395), which have not been introduced yet, to follow the logic. Couldn’t this entire section follow the section that starts at L. 397?

Thank you for the suggestion. We have moved the section regarding hybrid sterility to be after the viral piRNA section.

399: This is an awkward sentence. The potential antiviral role for piRNAs comes seemingly out of nowhere; why/how does this potential exist, or in what literature has this potential role been proposed? Also “silence” is rarely linked with the preposition “from.”  

We have modified the sentence to: “As described before, Drosophila piRNAs’ main function is to silence transposons which can damage the germline genome [34, 43, 117, 160, 174]. However, a few studies suggests an antiviral immunity role for Drosophila viral piRNAs seen in the OSS cell line

406: “that argue” to “who argue.”

Modified.

427: Did Fig. 1B come from virally-infected drosophilids?

We have updated the figure legend to say these profiles are from animals not infected with arboviruses.

454: “Very similar” -> “more similar in size and repeat content”

Modified.

470: I don’t understand this sentence, particularly how the prepositional phrase “inside or outside piRNA clusters that ultimately lead to the antiviral response” is intended to modify “such EVEs.”

We have modified the sentence for clarification: “An interesting question is how highly repetitive genomes (e.g. culicine mosquitoes) maintain antiviral EVEs located in either inside or outside piRNA clusters.”

489-490: “but perhaps…” this part of the sentence is weak. The following paragraph should be included in this paragraph.

We modified this paragraph to include the last paragraph. The following transition is as follows: “Why the exo-siRNA pathway has a dominant role in antiviral immunity is unknown. Insight into the host-virus evolution may explain why the endo-siRNA pathway has a smaller role against viruses. For example, the mosquito exo-siRNA pathway genes appear to undergo rapid, positive, and diversifying selection”

509: The tone here may not be what the senior authors are going for.

Lines 509-510 introduce many self-citations of questionable value based on the context.

We understand this language may be shaped by different cultural attitudes. We respectfully retain this text to reflect the ongoing collaboration between the Lau and Akbari labs.